# Thalamic T-Type Calcium Channels as Targets for Hypnotics and General Anesthetics

**DOI:** 10.3390/ijms23042349

**Published:** 2022-02-21

**Authors:** Tamara Timic Stamenic, Slobodan M. Todorovic

**Affiliations:** 1Department of Anesthesiology, Anschutz Medical Campus, University of Colorado, Aurora, CO 80045, USA; 2Neuroscience and Pharmacology Graduate Program, Anschutz Medical Campus, University of Colorado, Aurora, CO 80045, USA

**Keywords:** t-type calcium channels, general anesthesia, hypnosis, nonspecific thalamus, EEG recording, LFP recording

## Abstract

General anesthetics mainly act by modulating synaptic inhibition on the one hand (the potentiation of GABA transmission) or synaptic excitation on the other (the inhibition of NMDA receptors), but they can also have effects on numerous other proteins, receptors, and channels. The effects of general anesthetics on ion channels have been the subject of research since the publication of reports of direct actions of these drugs on ion channel proteins. In particular, there is considerable interest in T-type voltage-gated calcium channels that are abundantly expressed in the thalamus, where they control patterns of cellular excitability and thalamocortical oscillations during awake and sleep states. Here, we summarized and discussed our recent studies focused on the Ca_V_3.1 isoform of T-channels in the nonspecific thalamus (intralaminar and midline nuclei), which acts as a key hub through which natural sleep and general anesthesia are initiated. We used mouse genetics and in vivo and ex vivo electrophysiology to study the role of thalamic T-channels in hypnosis induced by a standard general anesthetic, isoflurane, as well as novel neuroactive steroids. From the results of this study, we conclude that Ca_V_3.1 channels contribute to thalamocortical oscillations during anesthetic-induced hypnosis, particularly the slow-frequency range of δ oscillations (0.5–4 Hz), by generating “window current” that contributes to the resting membrane potential. We posit that the role of the thalamic Ca_V_3.1 isoform of T-channels in the effects of various classes of general anesthetics warrants consideration.

## 1. Introduction

Since their discoveries, T-type calcium channels (T-channels) have been studied in the context of thalamocortical oscillatory behavior, synaptic plasticity, cell excitability, and involvement in rebound burst-firing [1,2,3]. T-channels were first described in the immature egg cell membrane of a starfish (*Mediaster aequalis),* where two distinct calcium currents were reported: one with activation at a membrane potential of −55/−50 mV (low-voltage-activated channels, LVA channels or T-channels) and the other at −7/−6 mV (high-voltage-activated or HVA channels) [4]. One very important property of T-channels is that they need only small depolarization to open, and they have the ability to form “window” currents around resting neuronal membrane potential (see Figure 1) [5]. In the thalamus, the occurrence of neuronal membrane potential bistability or destabilization is manifested as the transition of two resting membrane potentials (from “Down to Up states”). It is generally accepted this is most likely due to the “window” current generated by T-channels [5]. Neuronal membrane destabilization is crucial for generating rhythmic (slow/δ) oscillatory behavior during hyperpolarization states typically associated with sleep, sedation, hypnosis, or anesthesia. During neuronal hyperpolarization, T-channels are de-inactivated (recovered from the inactivation), allowing them to open after depolarization and trigger a low-threshold calcium spike (LTS) crowned with a barrage of action potentials (APs, rebound burst-firing pattern, Figure 1,for details see [6]). It is well-documented that during natural sleep or general anesthesia, inhibitory synaptic inputs hyperpolarize thalamic cells enough to recover T-channels from inactivation and consequently allow them to generate characteristic burst-firing and network oscillations. Importantly, it has been shown that in the burst-firing state, the thalamus does not conduct sensory information, a property crucial for natural sleep and general anesthesia effects [7].

There are three known isoforms of pore-forming α subunit of T-channels, known as α1G (CACNA1G, Ca_V_3.1), α1H (CACNA1H, Ca_V_3.2), and α1I (CACNA1I, Ca_V_3.3), which are expressed and localized in different brain regions, including the thalamus [8]. In most of the glutamatergic thalamic nuclei, the dominant subtype is the Ca_V_3.1 channel expressed on soma and dendrites. In the thalamic reticular nucleus (TRN), which is composed of GABAergic neurons, the most abundantly expressed are Ca_V_3.2 (mostly on cell somas) and Ca_V_3.3 (mostly on dendrites) T-channel isoforms [8]. Consistent with their essential roles in neuronal excitability, T-channel dysfunction has been implicated in epilepsy, sleep disorders, pain, neurological disorders, neuropsychiatric disorders, cognitive disorders, as well as in chronic thalamocortical hyperexcitability following exposure to general anesthesia during brain development [9,10,11,12,13].

One of the fundamental challenges in pharmacology remains to decipher mechanisms of action of general anesthetics. General anesthesia is a drug-induced, reversible condition composed of the behavioral states of hypnosis, amnesia, analgesia, and loss of motor reflexes (immobilization) during a surgical procedure. The mechanisms of loss of consciousness (hypnosis) have been studied for decades using neuroimaging [14,15]. An early theory proposed that the nonspecific alteration of the lipid membrane in nerve cells accounts for the anesthetic state [16,17]. However, advancements in research in the last three decades have disputed the nonspecific lipid membrane theory [18] and confirmed that general anesthetics act through multiple specific proteins on the neuronal membrane, and that different ion channels that control neuronal excitability may mediate their clinical effects. It is generally accepted that general anesthetics act on synaptic inhibition on the one hand (the potentiation of GABA-mediated transmission, e.g., propofol), or glutamate-mediated synaptic excitation on the other (the inhibition of NMDA receptors, e.g., ketamine, nitrous oxide), but they can also have effects on numerous other proteins, receptors, and channels [19,20,21,22]. It is known that general anesthetics markedly reduce the global cerebral metabolic rate and blood flow, and the thalamus appears to be a common site of modulation by several anesthetics [14,23,24]. The loss of consciousness induced by anesthetics represents the disruption of higher-order cortical information integration, and it has been shown that the posterior parietal–cingulate–precuneus region and the nonspecific thalamus have a critical role in maintaining the state of consciousness (for details, see [14]).

The brain arousal system involves the nonspecific thalamus (intralaminar and midline complex) [25]. It has been shown that the repetitive low-frequency stimulation of intralaminar thalamic nuclei is associated with sleep and drowsiness, while the high-frequency stimulation of the same area can desynchronize the cortex and elicit arousal [26]. The central nucleus of the thalamus (CeM) is a part of the rostral intralaminar complex with the projections to the cortex (anterior and posterior parts), the amygdala, the nucleus accumbens, the claustrum, the caudate–putamen, and the olfactory tubercle [27]. Some studies revealed that the nonspecific thalamus acts as a crucial center for brain network connectivity and is important for alterations induced by general anesthesia and natural sleep in rodents and humans [28,29,30,31]. The stimulation of the central thalamus (including the mediodorsal and intralaminar nucleus) in monkeys during the unconscious state caused the reversal of neurophysiological signs of the hypnotic state [32]. Consistent with this idea, microinjections of general anesthetics into the CeM region in rodents was shown to facilitate hypnosis, while injections of an antibody against voltage-gated potassium channels promoted arousal and the reversal of anesthetic-induced hypnosis [33,34]. Importantly, clinical studies have shown that deep brain stimulation of adjacent nuclei in the nonspecific central thalamus improved the state of consciousness in patients with severe brain injury [35]. Hence, this motivated us to investigate the role of the Ca_V_3.1 isoform of T-channels in regulating excitability states of the CeM and its role in neurosteroid-induced hypnosis and general anesthesia [36,37].

Interestingly, studies showed the evolutionary existence of a voltage-gated calcium current with the kinetic features of a T-channel current in *C. elegans*, *Drosophila*, and *Trichoplax adhaerens* [38,39]. Although there are reports that show the behavioral effects of volatile anesthetics in *C. elegans*, the potential involvement of T-channels in these effects was not investigated [40]. For these reasons, our studies focused on rodents with reference to humans and nonhuman primates.

## 2. Electroencephalographic Patterns during Unconsciousness and Anesthesia

Since the first description of electroencephalography (EEG), physiologically relevant rhythms have been investigated during different behavioral states and are classified as: δ (0.5–4 Hz), θ (4–8 Hz), α (8–13 Hz), β (13–30 Hz), low γ (30–50 Hz), and high γ (50–100 Hz) [3]. Sleep- and drug-induced unconsciousness share some similar mechanisms and involve the same brain regions [41,42]. For example, during natural sleep and under general anesthesia, the thalamocortical system can generate slow oscillations with high amplitudes, and conversely, during conscious states, distinct, faster oscillations are dominant [3,40,43]. The administration of a small dose of hypnotics, such as positive GABA_A_ receptor modulators (propofol, barbiturate, and etomidate), induces a state of sedation with paradoxical excitation and an increase in β activity [44]. During the maintenance period of general anesthesia in phase 1 (light state), a decrease in β and increase in α EEG activity is dominant, while in phase 2 (intermediate state), both δ and α oscillations increase similarly to slow-wave non-rapid-eye movement (NREM) sleep [44]. In phase 3 (deeper state), burst-suppression activity is observed, and in phase 4, a largely flat-isoelectric EEG pattern is seen [44].

### 2.1. Human Data

Many different classes of anesthetics exist within common clinical use, and those of note include volatile anesthetics (such as sevoflurane), GABA_A_ receptor modulators (propofol), and NMDA antagonists (ketamine and nitrous oxide). During sevoflurane-induced hypnosis, EEG and positron emission tomography (PET) analysis was used to identify changes in cerebral blood flow and metabolic activity in frontal, parietal, and thalamic regions [45]. While an increase in frontal β power occurs during sevoflurane-induced sedation and persists despite the loss of responsiveness, conversely δ, θ, and α band power remain unchanged (for a review, see [45]). It has been shown that frontal α power does not consistently emerge in deep hypnosis with sevoflurane but can change over the transition from wakefulness to sevoflurane-induced hypnosis [46,47,48,49]. With higher sevoflurane concentrations, α power increases and can consist of sleep-spindle-like activity during the maintenance phase of anesthesia [45]. On the other hand, specific EEG readout during sevoflurane-induced hypnosis has been shown to include the domination of δ waves and the existence of a burst-suppression pattern [45,49,50].

During propofol-induced hypnosis, coupling between low frequencies and amplitudes of α rhythms were observed with the dominant slow oscillations similar to sleep during unconsciousness in humans [51,52]. Additionally, it has been shown that in propofol-induced general anesthesia before the generation of a burst-suppression pattern, the main rhythm consists of α and δ band activity [53]. In humans with an implanted deep electrode, an increase in α and decrease in γ power were observed in both deep cortical (ACC, anterior cingulate cortex) and subcortical (sensory thalamus, periaqueductal gray) areas during propofol-induced hypnosis [54]. Moreover, hypnotic state under propofol was marked simultaneously by an increase in low-frequency EEG power (<1 Hz), the loss of spatially coherent occipital α oscillations, and the appearance of spatially coherent frontal α rhythms in humans [55].

Following ketamine-induced hypnosis in humans, a γ-burst EEG pattern was observed with alternating slow-δ and γ oscillations [56]. Additionally, ketamine hypnosis showed increased θ and decreased α/β oscillations [56].

Nitrous oxide (N_2_O) is usually used along with other anesthetics as an adjuvant agent as general anesthesia cannot be achieved with N_2_O alone under normobaric conditions. In healthy male volunteers, some studies reported a reduction in total EEG power with N_2_O and a paradoxical reduction in δ oscillations [57]. Others reported an increase in θ, β, and low and high γ band powers under sedative N_2_O concentration [58]. Interestingly, the administration of a high dose of N_2_O in combination with inhalation anesthetics (sevoflurane, desflurane, or isoflurane) was associated with dominant slow-δ oscillations [59]. The generation of these slow-δ rhythms could be due to the blockade of excitatory inputs (NMDA glutamate projections) from the brainstem to the thalamocortical neurons [59].

### 2.2. Animal Data

Similarly to human data, a low dose of sevoflurane in rats produced an increase in β/low γ power in the cortex and central thalamus [60]. During sevoflurane-induced loss of movement, β/low γ activity decreased with the generation of coherent slow-δ oscillations [60]. At higher doses, sevoflurane induced the loss of the righting reflex with the characteristic coherent slow-δ oscillations [60]. Additionally, the dominant slow waves and the burst-suppression pattern was observed with the higher isoflurane concentration in rats [61].

In rats, propofol-induced α oscillations synchronized between the thalamus and the medial prefrontal cortex, with the development of coherent thalamocortical δ oscillations at deep levels of hypnosis [62]. On the contrary, a sub-anesthetic dose of ketamine, an NMDA antagonist, induced robust, spontaneous γ oscillations that were initially modulated by slow oscillations (0.3 Hz) in monkeys [63]. In rodents, it has been shown that ketamine increased power in the 30–50 Hz frequency band (low γ) with or without a rise in high γ oscillations [64,65].

### 2.3. Summary

Slow-δ and α oscillations are dominant EEG signatures for propofol-induced anesthesia, which is a finding consistent with EEG patterns observed with other intravenous anesthetics targeting GABA_A_ receptors (for details, see [66]). Similarly, during sevoflurane-induced anesthesia, slow-δ, θ, and α oscillations are predominant EEG signatures, consistent with other volatile general anesthetics (desflurane and isoflurane) [66]. The similarities between the EEG patterns of GABA_A_-receptor-targeting anesthetics, propofol, and volatile general anesthetics seem to include the enhancement of GABA_A_-receptor-mediated inhibitory postsynaptic currents [66]. On the contrary, low γ oscillations that are interspersed with slow-δ oscillations are the predominant EEG signatures of general anesthesia maintained with ketamine, a prototypical injectable general anesthetic and an NMDA receptor antagonist [66]. It is worth mentioning that these changes are dose- and state-dependent for each anesthetic/hypnotic agent.

## 3. T-Channels in Sleep, Hypnosis, and Anesthesia

It is well-known that T-type channels play a number of different and essential roles in neuronal oscillation generated by thalamic neurons during NREM sleep [67,68]. Additionally, the thalamus is required for slow-wave frequency tuning, which is highly dependent on T-channels, during both NREM sleep and anesthesia [68,69]. It has been shown that sleep behavior was not changed in animals that lacked a Cav3.1 channel in cortical pyramidal neurons [70]. In contrast, the focal deletion of the Ca_V_3.1 channel in the rostral–midline thalamus caused frequent and prolonged arousal, with fragmented and reduced sleep [70]. These results support the hypothesis that thalamic T-channels play essential roles in sleep stabilization. Other studies demonstrated that mice lacking the Ca_V_3.1 isoform of T-channels exhibited a reduction in thalamic δ oscillations and sleep spindles during urethane- and barbiturate-induced hypnosis [71,72]. Additionally, Ca_V_3.1 KO mice under ketamine and ethanol administration displayed attenuated cortical and mediodorsal thalamic low-frequency oscillations (1–4 Hz) when compared to WT mice [73]. Towards this end, it has been previously shown that Ca_V_3.1 KO mice did not have different ED_50_ for the loss of the righting reflex caused by volatile anesthetics (isoflurane and halothane), but these mutant mice had significantly delayed onset of anesthetic induction, as measured by the time to the loss of the righting reflex [74]. Surprisingly, in the same study, authors reported that the duration of the loss of the righting reflex and onset of anesthetic induction induced by injections of propofol was not different between WT and Ca_V_3.1 KO mice. This suggests that different classes of anesthetics may have different mechanisms of interactions with Ca_V_3.1 channels and other molecular targets in the thalamocortical circuitry. In agreement with these findings, our group reported that global Ca_V_3.2 KO mice did not have different ED_50_ for the loss of the righting reflex caused by volatile anesthetics (isoflurane) but had significantly delayed onsets of anesthetic induction with isoflurane and not with propofol [75]. In contrast to these studies, the global deletion of the Ca_V_3.3 isoform of T-channels in mice per se did not affect the ED_50_ for the loss of the righting reflex, but facilitated anesthetic induction caused by isoflurane [76]. These findings point to the different roles of T-channel isoforms in anesthetic-induced hypnosis.

In vivo animal electrophysiological data showed that anesthetics have the ability to modify thalamocortical signaling by altering the neuronal firing patterns of the thalamic network at the cellular level by hyperpolarizing resting membrane potentials of thalamic neurons [77,78]. It is known that most thalamic cells have prominent bursting and not tonic activity during the administration of anesthetics, similar to different sleep states [77,79]. It has been hypothesized that anesthetics may cause hypnosis because of the hyperpolarization-induced blockade of the thalamocortical cells and networks essential for consciousness [77,80]. It has been shown that the thalamus can generate spindle oscillations (7–14 Hz) at resting membrane potentials (RMPs) of −60 mV and produce a slower δ rhythm consisting of low-threshold spikes (LTS) followed by afterhyperpolarizing potentials during natural sleep and hypnosis/anesthesia at a more negative RMP (<−65 mV) [81,82]. Previously, it was thought that for the generation of thalamic δ oscillations during the hyperpolarization of the neuronal membrane, both a hyperpolarization-activated (Ih) current and T-current are required [81]. However, a recent paper showed the dominant role of T-currents in δ rhythm generation and the supporting role of Ih currents in the amplification and stabilization of δ oscillations [83].

### 3.1. Role of the Ca_V_3.1 Isoform of T-Channels in Anesthesia Induced by Isoflurane

As mentioned earlier, studies with mice lacking Ca_V_3.1 channels showed that the mutant animals exhibited delayed induction with volatile anesthetics, including isoflurane [74]. In addition, we and others revealed that Ca_V_3.1 T-channels are inhibited by prototypical volatile anesthetic isoflurane in both the sensory thalamus (TRN, VB) and the nonspecific thalamus (CeM) at clinically relevant concentrations [37,84,85,86]. Previous studies demonstrated the role of the CeM as the neuroanatomic site in mediating arousal response during anesthetic administration in rats [28,33,34,87]. Our data confirmed that under isoflurane anesthesia, neuronal hyperpolarization is capable of removing T-channels from inactivation (de-inactivation), and the generation of bursting and oscillatory behavior within the thalamocortical loop is facilitated [37]. Not only did isoflurane hyperpolarize CeM neurons, but it inhibited both tonic and rebound burst-firing in the CeM neurons in WT, but not Ca_V_3.1 null animals. Similarly to previous studies regarding VB thalamic neurons [88], while tonic firing was spared, CeM neurons from Ca_V_3.1 KO animals did not show rebound burst activity. Our recordings of local field potentials (LFPs) from the CeM confirmed that under isoflurane-induced hypnosis, δ-frequency oscillations increased in WT mice but not in Ca_V_3.1 KO animals. The lack an increase in δ activity could be partially because of the inability of isoflurane to hyperpolarize CeM neurons in mutant animals and generate slow/δ oscillations ([37], Figure 2).

Previous studies reported that during NREM sleep, cortical power densities in low frequencies, including the δ band, were decreased in Ca_V_3.1 KO mice [71], but that the cortical spindles were not altered [72]. While in our study, we did not investigate the spindle component, we found increased power density in the spindle-like frequency range (8–13 Hz) in the CeM in mice lacking a Ca_V_3.1 channel during the quiet awake state and under 1 vol%, but not under 2 vol% of isoflurane anesthesia [37]. However, note that α oscillations induced by general anesthetics occur in a frequency range and spatial distribution similar to sleep spindles (12–16 Hz), although there are important differences between these two rhythms; for details, see [66].

Most clinically used general anesthetics and some metabolic abnormalities (profound acidosis and hypercapnia) can induce a characteristic EEG pattern with a burst (high-amplitude oscillations) and suppression (silenced cortical activity) phase. In our study, the suppression to burst ratio (BSR) was higher during 1.4 vol% of isoflurane administration in Ca_V_3.1 KO animals than in control mice, indicating greater thalamocortical suppression in the mutants (Figure 2, Table 1). Moreover, during 2 vol% isoflurane administration, the main observed thalamic rhythm was in the δ frequency range during suppression mode in the control, but not in the Ca_V_3.1 KO mice [37]. Notably, this was not seen in cortical EEG recordings. It is important to note that the difference between mutant and WT animals in terms of the BSR disappeared with the higher isoflurane concentration (Table 1), which suggests that different targets experience different effects at higher anesthetic concentrations. As all anesthetics are promiscuous drugs, the direction and magnitude of response in Ca_V_3.1 KO mice may depend on the effects of that anesthetic on other targets, including possible compensatory changes that exist in mutant animals.

### 3.2. Role of the Ca_V_3.1 Isoform of T-Channels in Nonspecific Thalamus in Hypnosis Induced by Neuroactive Steroids

The idea that neuroactive steroids can have sedative/hypnotic properties has been around since the introduction of alphaxalone [(3α,5α)3-hydroxypregnane−11,20-dione]. The neurosteroid analogs are potent GABA modulators that potentiate postsynaptic GABA_A_ currents, as well as inhibitors of voltage-gated calcium channels (reviewed in [89]). Neuroactive steroids mostly belonging to pregnane and androstane groups, such as the mixture of alphaxalone and alphadolone (Althesin^®^), were prepared for clinical use as anesthetics in the 1970s, but due to the high incidence of anaphylactic reactions, they were withdrawn from clinical use [90,91]. Today, alphaxalone is still used as an effective anesthetic in veterinary medicine (Alfaxan^®^) and is currently undergoing clinical trials in humans (Phaxan^®^) [92]. The ongoing use of alphaxalone continues to encourage the future development of synthetic neurosteroids with hypnotic/anesthetic properties.

The ability of general anesthetics to induce the safe and reversible loss of consciousness is of paramount importance; however, recent data from in vivo animal models have suggested that most commonly used general anesthetics are neurotoxic (i.e., causing neuronal apoptosis) to the developing mammalian brain and are implicated in causing cognitive deficits later in life. Thus, further research into cellular mechanisms of action of currently available anesthetics and the development of novel classes of general anesthetics for clinical practice with reduced neurotoxicity is warranted. We recently reported that the application of a neurosteroid analog (3β,5β,17β)-3-hydroxyandrostane-17-carbonitrile (3β-OH) in rats and rat pups produces a hypnotic effect without activating neuronal apoptosis, unlike other general anesthetics [93,94]. In addition, we investigated the role of 3β-OH on neuronal T-channels in the thalamus (TRN, CeM) and in sensory neurons of the dorsal root ganglion (DRG) [36,95,96]. We found that 3β-OH, although it does not act on GABA_A_ receptors, it can still effectively induce hypnosis in rat pups [94] and adult mice [36]. This suggests that 3β-reduced neurosteroids produce their hypnotic effects through alternate mechanisms, partially as potent inhibitors of T-channels [36,96,97]. We showed that the Ca_V_3.1 isoform is important for the inhibitory effect of 3β-OH on the CeM neuronal excitability [36]. Additionally, we confirmed that Ca_V_3.1 T-channels are important for the 3β-OH-induced hypnotic development in vivo by measuring the loss of the righting reflex in WT and mutant Ca_V_3.1 KO animals. Animals that lacked the Ca_V_3.1 T-channel isoform had several-fold shorter durations and lower calculated median effective doses for the loss of the righting reflex in comparison to control (WT) animals [36].

Bearing in mind the role of the Ca_V_3.1 isoform in the generation of thalamic δ oscillations and sleep spindles during hypnosis and anesthesia, a reduction in total δ, θ and α power in Ca_V_3.1 KO mice under a hypnotic dose (80 mg/kg) of 3β-OH was expected [36]. Similarly to the isoflurane effect in CeM neurons, 3β-OH did not hyperpolarize CeM neurons in the mutant mice, and a lack of 3β-OH-induced hyperpolarization can explain both the differences in low-frequency oscillations between WT and Ca_V_3.1 KO mice, and also the lack of effect on LORR in the Ca_V_3.1 null mice [36]. Furthermore, our findings revealed for the first time that injection of 3β-OH at 80 mg/kg induced a hypnotic state in WT mice (Figure 3), but was insufficient to do so in Ca_V_3.1 KO mice [36]. The injection of a higher 3β-OH dose (120 mg/kg) induced an anesthesia state with the characteristic transient rise in β-frequency oscillations during induction and a burst-suppression pattern at later time points, suggesting deeper thalamocortical inhibition (Figure 3). A sex-dependent effect was reported regarding neuroactive steroids, which had a more pronounced effect in female animals, and in Figure 3, representative heat maps show greater suppression in female mice with both the hypnotic and anesthetic dose of 3β-OH. Similar sex differences with 3β-OH were previously reported in rats (for details, see [93]).

In our previous studies, we established the role of voltage-gated calcium channels in antinociceptive effects of an endogenous 5β-reduced neuroactive steroid molecule epipregnanolone (EpiP, [(3β,5β)-3-hydroxypregnan-20-one]) in rats and mice [97,98]. Similarly to 3β-OH, we demonstrated that EpiP blocks T-type calcium channels in sensory neurons without having an effect on GABA_A_ currents [97,99,100]. Interestingly, we demonstrated that T-channel isoforms contribute differently to EpiP-induced hypnosis in mice [101]. We found that EpiP is an effective dose-dependent hypnotic when given alone and that it significantly lowers the isoflurane and sevoflurane concentration required to induce immobility and hypnosis [101]. Additionally, after systemic EpiP injection in WT mice, we observed a rise in total power in all EEG frequencies [101]. Similarly to changes observed using the other known sedative/hypnotic drugs, with the administration of EpiP, we detected a rise in relative δ and β power and a drop in relative γ power 30 min after EpiP administration. It has been shown that many different classes of general anesthetics first induce sedation/hypnosis with a characteristic rise in β oscillation followed by a rise in δ oscillations, which are dominant during deeper levels of anesthesia [66]. Consistent with this idea, we found that Ca_V_3.1 KO mice, but not Ca_V_3.2 and Ca_V_3.3 KO mice, exhibited resistance to EpiP-induced hypnosis, as demonstrated by a shorter loss of righting reflex duration and a significant reduction in δ oscillations when compared to the control WT animals [101]. These data confirm results with 3β-OH, the synthetic neurosteroid analog of EpiP, which showed a similar EEG signature to EpiP during hypnosis induced in WT animals [36]. Interestingly, in the same study, we showed that when compared to WT mice, the onset of EpiP-induced hypnosis was delayed in Ca_V_3.2 KO mice, but not in Ca_V_3.1 and Ca_V_3.3 KO mice. However, we observed that among all three T-channel isoforms, Ca_V_3.1 had the greatest relevance with regard to EpiP-induced hypnotic effects. We speculate that the distinct hypnotic effects of EpiP and isoflurane across all three T-channel isoforms are due to their differential expression in thalamocortical circuitry.

## 4. Conclusions

It is well established that the loss of consciousness under anesthesia is associated with an increase in δ oscillations and that T-channels play an important role in δ oscillation generation [60,61,62]. It was demonstrated that a lack of rebound bursting and the inability of anesthetics/hypnotics to hyperpolarize neurons and to increase slow oscillations in Ca_V_3.1 KO mice can likely explain their inability to induce hypnosis in mutant mice. Our data are consistent with idea that the inability of anesthetics to hyperpolarize CeM neurons in mutant mice is due to lack of a T-channel-dependent “window current”. At rest, a T-type “window current” provides a steady influx of calcium ions near resting membrane potentials and, in turn, depolarizes the neuronal membrane. In contrast, blocking a T-channel-dependent “window current” causes neuronal hyperpolarization. The central nucleus of the thalamus, as a part of the nonspecific thalamus, acts as a key hub in brain network connectivity alterations induced by general anesthesia and natural sleep in rodents and humans. It has been shown that central thalamic stimulation during the hypnotic state can reverse the neurophysiological signs of the unconsciousness. Due to the abundant expression of Cav3.1 T-channels in the nonspecific thalamus and their regulation of its excitability, we propose that the effects of various general anesthetics on thalamic Ca_V_3.1 channels warrant consideration.

## Figures and Tables

**Figure 1 ijms-23-02349-f001:**
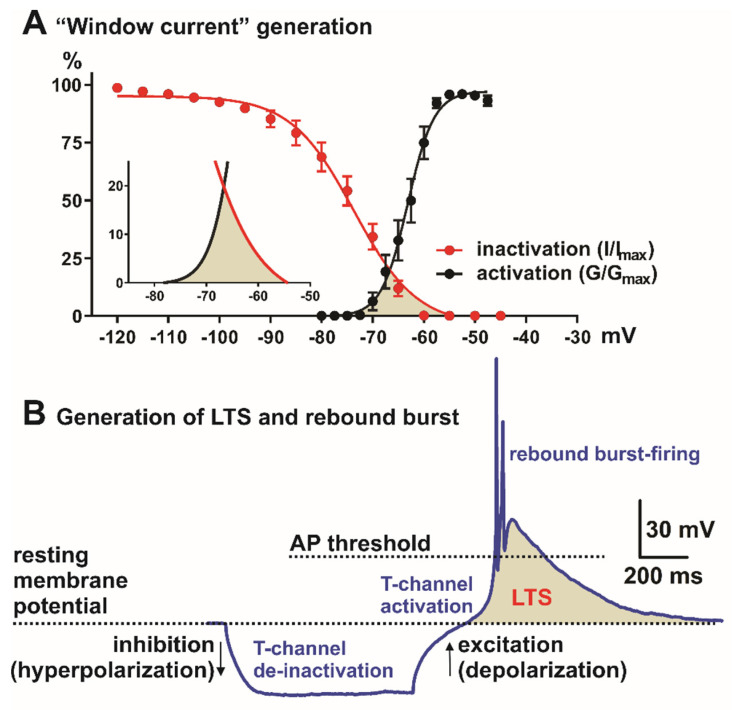
Neuronal Tchannel properties in the nonspecific thalamic nucleus (CeM). (**A**) “Window” current generation, red trace—steady-state inactivation and black—steady-state activation T-channel kinetics (from voltage-clamp experiments). Inset in A is enlarged “window” current (shaded area) generated by overlap between steady-state activation (black) and inactivation (red) curves. (**B**) Role of the T-channels in LTS (low-threshold spike, shaded area) and rebound burst generation. Note that in most systems, LTS and burst-firing cannot be generated from the resting membrane potential, but neurons need hyperpolarization of the cell membrane in order to allow T-channel de-inactivation (recovery from inactivation). AP—action potential.

**Figure 2 ijms-23-02349-f002:**
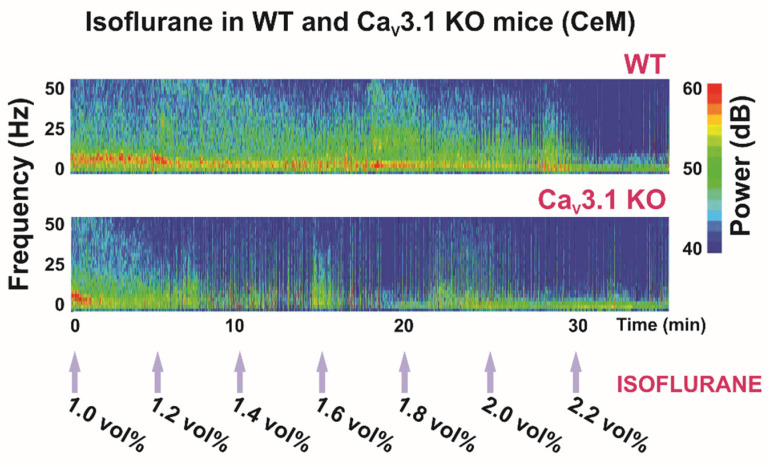
Thalamic (central medial nucleus of thalamus—CeM) heat maps of local field potentials (LFPs) under isoflurane in WT and Ca_V_3.1 KO animals.

**Figure 3 ijms-23-02349-f003:**
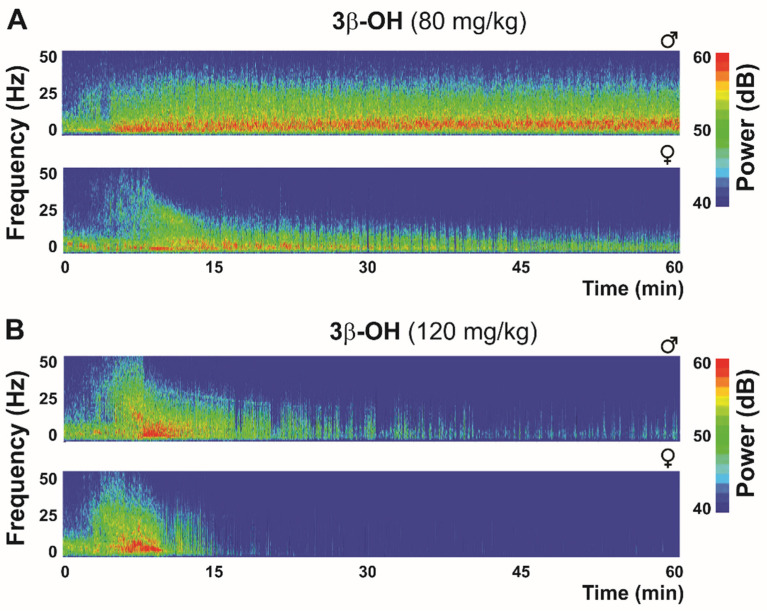
Thalamic (central medial nucleus of thalamus—CeM) heat maps after hypnotic (80 mg/kg (**A**)) and anesthetic (120 mg/kg (**B**)) dose of neuroactive steroid 3β-OH in WT mice.

**Table 1 ijms-23-02349-t001:** Suppression to burst ratio (BSR) in WT and Ca_V_3.1 KO animals under isoflurane (mean ± SEM).

Isoflurane Concentration	WT	Ca_V_3.1 KO
1.4 vol%	0.10 ± 0.03	0.38 ± 0.07 ^1^
1.6 vol%	0.37 ± 0.11	0.52 ± 0.06
1.8 vol%	0.64 ± 0.08	0.68 ± 0.05
2.0 vol%	0.78 ± 0.04	0.75 ± 0.05

^1^ Statistically significant WT vs. Ca_V_3.1 KO mice (*p* < 0.05).

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
