# Peer review of "Thalamic T-Type Calcium Channels as Targets for Hypnotics and General Anesthetics"

_ijms, 2022, doi:10.3390/ijms23042349_

Round 1

Reviewer 1 Report

This is a very well thought out manuscript which describes the implication of T-type calcium channels (especially the CaV3 T-channel isoform) and their involvement in the mechanism of general anesthesia. The manuscript would benefit from some more formal editing of English grammar/syntax in many areas as the wording often comes across either incorrectly or awkwardly. Additionally, there were numerous misspellings that should be readily amenable to a simple Spellchecker run. The manuscript could be worthy of publication after a few additional issues/questions can be addressed as noted below:

page 1 line 29: I find the word "deinactivated" rather confusing. Doesn't this also merely mean activated? Is there greater explanation needed here?

On page 2, the authors present a very nice overview of the current understanding of anesthetic mechanisms and the importance of thalamic nuclei. This motivated their interest in the T-channels in nonspecific thalamic nuclei. However, given the similar effects of anesthetics across wide ranges of evolutionary phylogeny, can the authors comment on the presence or absence of these channels in organsism which respond to anesthetics but which do not have a thalamus or related extrathalamic nuclei? Are the extrathalamic nuclei akin to other areas in more primitive organisms?

page 2 line 84, I think electroencephalograhic should be electroencephalograhy.

On page 3, the authors present a very nice overview of various general anesthetics (acting through either GABA or NMDA receptors). Their argument would be strengthened further by a mention of the effects of nitrous oxide on NMDA receptors. Is this latter effect consistent with their other statements?

page 4, line 151, dependable should be dependent

page 4, line 164, frequenty is misspelled and should be frequency

page 4, line 190, Previosly  and tought are misspelled and should be Previously and thought, respectively.

page 5, line 207, unefected is misspelled and should be uneffected

page 5, figure 1 (and most other figures for that matter) require larger size fonts for labels to be readable without magnfication on a large screen.

page 5 table one: Why does the difference between WT and KO go nearly to zero as the isoflurane concentration goes up? Are we seeing effects across larger populations of different receptors at higher concentations of volatile anesthetics which are known to have effects at multiple sites?

page 6: Alphaxalone is not only available for veterinary use as Alfaxan but is currently undergoing clinical trials in humans as Phaxan and marketed by Drawbridge pharmaceuticals. It is also reformulated in a cyclodextrin.

page 7 line 295, grater should be greater

Generally speaking, why does it seem that some classes of anesthetics work better/quicker in the T channel KO's while the effects of other anesthetics seem to be curtailed? Do the authors believe that, as noted for the isoform KO effects on epipregnanolone, other anesthetics have subtype isoform specific dependencies?

Author Response

This is a very well thought out manuscript which describes the implication of T-type calcium channels (especially the CaV3 T-channel isoform) and their involvement in the mechanism of general anesthesia. The manuscript would benefit from some more formal editing of English grammar/syntax in many areas as the wording often comes across either incorrectly or awkwardly. Additionally, there were numerous misspellings that should be readily amenable to a simple Spellchecker run. The manuscript could be worthy of publication after a few additional issues/questions can be addressed as noted below:

page 1 line 29: I find the word "deinactivated" rather confusing. Doesn't this also merely mean activated? Is there greater explanation needed here? This term is commonly used in the field of voltage-gated calcium channels and it refers to “recovery from inactivation”. We explained this in the revised text.

On page 2, the authors present a very nice overview of the current understanding of anesthetic mechanisms and the importance of thalamic nuclei. This motivated their interest in the T-channels in nonspecific thalamic nuclei. However, given the similar effects of anesthetics across wide ranges of evolutionary phylogeny, can the authors comment on the presence or absence of these channels in organsism which respond to anesthetics but which do not have a thalamus or related extrathalamic nuclei? Are the extrathalamic nuclei akin to other areas in more primitive organisms? This is an interesting question. We addressed this in the text.

page 2 line 84, I think electroencephalograhic should be electroencephalograhy. Thank you for your comment. We changed this.

On page 3, the authors present a very nice overview of various general anesthetics (acting through either GABA or NMDA receptors). Their argument would be strengthened further by a mention of the effects of nitrous oxide on NMDA receptors. Is this latter effect consistent with their other statements? We agree with this comment and we discussed effects of the N2O in the text.

page 4, line 151, dependable should be dependent. We changed this.

page 4, line 164, frequenty is misspelled and should be frequency. We changed this.

page 4, line 190, Previosly  and tought are misspelled and should be Previously and thought, respectively. We changed this.

page 5, line 207, unefected is misspelled and should be unaffected. We changed this.

page 5, figure 1 (and most other figures for that matter) require larger size fonts for labels to be readable without magnfication on a large screen. We changed Figures and enlarged size fonts on Figures.

page 5 table one: Why does the difference between WT and KO go nearly to zero as the isoflurane concentration goes up? Are we seeing effects across larger populations of different receptors at higher concentations of volatile anesthetics which are known to have effects at multiple sites? We agree with this statement and we have now included this explanation in the revised manuscript.

page 6: Alphaxalone is not only available for veterinary use as Alfaxan but is currently undergoing clinical trials in humans as Phaxan and marketed by Drawbridge pharmaceuticals. It is also reformulated in a cyclodextrin. We agree and we added this in the manuscript.

page 7 line 295, grater should be greater. We changed this.

Generally speaking, why does it seem that some classes of anesthetics work better/quicker in the T channel KO's while the effects of other anesthetics seem to be curtailed? Do the authors believe that, as noted for the isoform KO effects on epipregnanolone, other anesthetics have subtype isoform specific dependencies? We agree that this is somewhat controversial. We believe that any deviation of the response to a particular drug in KO mice when compared to the WT mice validates importance of that particular channel in question. However, since all anesthetics are promiscuous, direction and magnitude of response in KO mice may depend on the effects of that anesthetic on other targets, including possible compensatory changes. This issue is discussed in the revised manuscript.

Note that changes in the manuscript are in blue font.

Reviewer 2 Report

The significance of the topic discussed is clearly high, however, it seems that the paper is troubled with multiple major and minor issues. Although the authors are well-known in the field of anesthesiology, electrophysiological terminology was not used properly and the manuscript failed to show clear understanding of anesthesia-related electrophysiological phenomena. The review is unfocused in terms of the chosen topic and lacks clear and useful conclusions. Please see the comments below.

Abstract

Abstract lacks specificity and should be extended. What are the main conclusions/highlights in this review?

“Specifically, we focused on the nonspecific”, the statement is unnecessarily confusing and needs revision.

Introduction

“is need for smaller depolarization”. Smaller relatively to what? Action potential?

“They are crucial for the neuronal membrane destabilization”. What does “membrane destabilization” mean?

“generation of rhythmic oscillatory behavior during hyperpolarization states (sleep, sedation, hypnosis or anesthesia).” The type of behavior discussed needs defining; what type of behavior is being referred to?

“hyperpolarization states” of which neurons specifically? Are interneurons included in this sentence?

“During neuronal hyperpolarization T-channels are deinactivated, so they can open after depolarization”. Does “deinactivated” mean “activated”? Does “after depolarization” mean “repolarization”? In my experience, I have seen T-channels to be described as functionally inactive during hyperpolarization and active during depolarization. Please clarify, if this is not the case.

  1. Electroencephalographic patterns during unconsciousness and anesthesia

 “Administration of a small dose of hypnotics, such as positive GABAA receptor modulators”. Examples of pure positive GABAA receptor modulators would be helpful to illustrate the mechanisms.

“Following the ketamine-induced unconsciousness in humans y-burst EEG pattern was observed with the altering slow-δ and y oscillations”. Actually, the y-burst EEG pattern consisted of alternating delta and gamma oscillations. This should be corrected.

Overall, section 2.1 provides a variety of information about effects of anesthesia, but it is not clear how everything is connected to the main topic of the paper.

2.3. Summary

The described effects are dose-dependent, not universal. This should be clarified.

“others” – Line 141

  1. T-channels in sleep, hypnosis and anesthesia

“thalamic T-channels are essential for arousal and sleep stabilization”. What about other channels in the same thalamic neurons; are they essential too? It would be interesting to discuss this question.

“This suggests that different classes of anesthetics may have different mechanisms of interactions with CaV3.1 channels and other molecular targets in the thalamocortical circuitry.” Again, it would be useful to discuss how can we differentiate the effect on  CaV3.1 channels and other concurrent effects.

“The in vivo animal electrophysiological data showed that anesthetics have ability to modify thalamocortical signaling by altering the neuronal firing patterns of thalamic  network at the cellular level by hyperpolarizing resting membrane potentials of thalamic neurons [68,69]. It is known that most of the thalamic cells have prominent bursting and not tonic activity during admistration of anesthetics, similar as under different sleep states [68,70].” So, is it hyperpolarization or bursting?

“admistration” – Line 183

“non-secific” -  Line 199

“Our data confirmed that the CaV3.1 isoform of T-channels plays an important role in regulation of excitability and rhythmic activity of CeM during isoflurane-induced anesthesia”. What is the purpose of this regulation under anesthesia?

Figure 1. Was the frequency artificially cut off at approximately 40 Hz in the upper panel? The imaginary line of the cut off looks artificially straight. If it was not cut off, why has absolutely everything above 40Hz disappeared?

“While in our study we did not investigated” – line 220

“While in our study we did not investigated the spindle component, we found increased power density in spindle-like frequency range (8–13 Hz)”. Although it is typically 11-16Hz for spindles, perhaps you can say this, however, does it mean that your data contradict the data in [63]? If this is the case, it should be discussed in greater detail.

“in mice lacking CaV3.1 channel during quiet awake state and under 1 vol%, but not under 2 vol% of isoflurane anesthesia in CeM”. 2% isoflurane for mice is more than 1 MAC and you describe burst suppression in the next paragraph under 1.4%. What changes would you expect to see in 8-13Hz under 2%?

“Moreover, we found that during inhalation of 2 vol% isoflurane, dominant activity was δ oscillations during suppression mode in the WT mice, but not in the CaV3.1 KO animals”. What was the dominant activity in CaV3.1 KO animals if they have the same suppression to burst ratio as in WT group according to Table 1?

Figure 2 should be accompanied by group data with error bars if you want to show such a big difference between males and females.

“3.3. Role of the CaV3.3 T-channels in anesthesia induced by isoflurane” It is not clear why only isoflurane was chosen for this discussion. What about sevoflurane and other anesthetics?

“The TRN is anatomically positioned as an optimal site to monitor cortical sensory processing and the regulation of the thalamus [90].” This is arguable. Physiologically it sounds very unusual that thalamus monitors cortical processing. I do not understand the mechanism. Activation of the cerebral cortex typically suppresses activity in the thalamus. Additional support/explanaition is advised for this statement.

“It has been shown that the local optogenetical TRN activation induces slow wave activity similar to those seen in sleep [91]. Also, during the TRN stimulation animals exhibited behavioral changes consistent with a decrease in arousal state [91].” These studies do not prove the point that thalamus monitors cortical processing. They only show the effect of strong artificial stimulation of TRN on the cortex.

“The TRN possess an intrinsic firing ability mediated by CaV3.3 channels expressed predominantly in the dendritic branches that facilitate characteristic bursting induced by T-channel activation during sleep and anesthesia and tonic firing during awake state [89,92].” It would be interesting to discuss the driver for the spontaneous (?) activation of T-channels under these conditions.

Overall, sections 3.3 and 3.4 seems unfocused (as well as some other sections). They list different studies and results, but it is not clear what the implications are of the findings presented, especially in reference to the purpose of the manuscript.

The conclusions are descriptive and weak.

Author Response

The significance of the topic discussed is clearly high, however, it seems that the paper is troubled with multiple major and minor issues. Although the authors are well-known in the field of anesthesiology, electrophysiological terminology was not used properly and the manuscript failed to show clear understanding of anesthesia-related electrophysiological phenomena. The review is unfocused in terms of the chosen topic and lacks clear and useful conclusions. Please see the comments below.

We thank this reviewer for carefully reading our manuscript and raising many interesting questions. We have addressed most questions except those that are not currently well studied. Although our work and work of others have established that T-channels are crucial for characteristic neuronal oscillations in the thalamus during sleep and anesthesia (e.g. Franks 2008, Crunelli group), T- type channels do not fit a typical simplistic view that during anesthesia everything is inhibited. Hence, more work is needed to decipher their precise role in anesthesia. Thank you for reading our revision!

Abstract

Abstract lacks specificity and should be extended. What are the main conclusions/highlights in this review?

“Specifically, we focused on the nonspecific”, the statement is unnecessarily confusing and needs revision. We changed and extended abstract.

Introduction

“is need for smaller depolarization”. Smaller relatively to what? Action potential? The T-type channels are known to operate in subthreshold range, hence “small depolarization” is appropriate. We added detailed explanation in the manuscript.

“They are crucial for the neuronal membrane destabilization”. What does “membrane destabilization” mean? We imply “state that can transition from the resting membrane potential (thus stable) to action potential firing (unstable)”. We added detailed explanation in the manuscript.

“generation of rhythmic oscillatory behavior during hyperpolarization states (sleep, sedation, hypnosis or anesthesia).” The type of behavior discussed needs defining; what type of behavior is being referred to? We agree that this can be confusing for the general audience and we added detailed explanation in the manuscript.

“hyperpolarization states” of which neurons specifically? Are interneurons included in this sentence? We are referring to hyperpolarization of the thalamic cells which are dominantly glutamatergic with the exception of TRN which is composed from GABAergic neurons. The most of the neurons in the thalamus (and the brain) are under inhibition (“hyperpolarization state”) during sleep and general anesthesia. We edited the manuscript to address this.

“During neuronal hyperpolarization T-channels are deinactivated, so they can open after depolarization”. Does “deinactivated” mean “activated”? Does “after depolarization” mean “repolarization”? In my experience, I have seen T-channels to be described as functionally inactive during hyperpolarization and active during depolarization. Please clarify, if this is not the case. Actually, T-channels are one the few classes of ion channels (in addition to Ih) that are quite active during hyperpolarization of neuronal membrane. In order for T-channels to activate they need to recover from inactivation (de-inactivation) during hyperpolarization of the membrane such as those generated during inhibitory postsynaptic potentials (IPSPs). We clarified this in the text.

2. Electroencephalographic patterns during unconsciousness and anesthesia

 “Administration of a small dose of hypnotics, such as positive GABAA receptor modulators”. Examples of pure positive GABAA receptor modulators would be helpful to illustrate the mechanisms. We agree and we have added examples of the hypnotic drugs that are positive GABAA receptor modulators.

“Following the ketamine-induced unconsciousness in humans y-burst EEG pattern was observed with the altering slow-δ and y oscillations”. Actually, the y-burst EEG pattern consisted of alternating delta and gamma oscillations. This should be corrected. From the cited paper (Akeju, O.; Song, A.H.; Hamilos, A.E.; Pavone, K.J.; Flores, F.J.; Brown, E.N.; Purdon, P.L. Electroencephalogram signatures of ketamine anesthesia-induced unconsciousness. Clin. Neurophysiol. 2016, 127, 2414–2422, doi:10.1016/J.CLINPH.2016.03.005): “ Results: Following the administration of a bolus dose of ketamine to induce unconsciousness, we observed a “gamma burst” EEG pattern that consisted of alternating slow-delta (0.1-4 Hz) and gamma (~27-40 Hz) oscillations.”  Delta oscillations are usually from 0.5-4 Hz and slow oscillations are <1 Hz, however since both slow and delta oscillations can contain 1-4 Hz spectral power and can be distinguished based on their waveforms, it is common to analyze both of them as slow-delta oscillations (0.1-4 Hz).

Overall, section 2.1 provides a variety of information about effects of anesthesia, but it is not clear how everything is connected to the main topic of the paper. This section describes EEG effects of the general anesthetics in humans.

2.3. Summary

The described effects are dose-dependent, not universal. This should be clarified. We agree and we clarified this in the manuscript.

“others” – Line 141 We changed this.

3. T-channels in sleep, hypnosis and anesthesia “thalamic T-channels are essential for arousal and sleep stabilization”. What about other channels in the same thalamic neurons; are they essential too? It would be interesting to discuss this question. This is very interesting question. However since our review is about T-channels additional discussion about other channel contribution is out of the scope of this manuscript.

“This suggests that different classes of anesthetics may have different mechanisms of interactions with CaV3.1 channels and other molecular targets in the thalamocortical circuitry.” Again, it would be useful to discuss how can we differentiate the effect on  CaV3.1 channels and other concurrent effects. Indeed, it is experimentally very difficult to differentiate effect achieved just on a single type of the channel. We, and other researchers, used global KO, tissue specific knock down (KD) or CRE-specific KD animals in studies.

“The in vivo animal electrophysiological data showed that anesthetics have ability to modify thalamocortical signaling by altering the neuronal firing patterns of thalamic network at the cellular level by hyperpolarizing resting membrane potentials of thalamic neurons [68,69]. It is known that most of the thalamic cells have prominent bursting and not tonic activity during administration of anesthetics, similar as under different sleep states [68,70].” So, is it hyperpolarization or bursting? Neurons that express T-channels typically have burst-firing mode after hyperpolarization of the membrane (e.g. after IPSPs), rarely they can generate burst-firing from the resting membrane potential at which most T-channels are inactivated. We explained this in the manuscript.

“admistration” – Line 183 We changed this.

“non-secific” -  Line 199 We changed this.

“Our data confirmed that the CaV3.1 isoform of T-channels plays an important role in regulation of excitability and rhythmic activity of CeM during isoflurane-induced anesthesia”. What is the purpose of this regulation under anesthesia? Since T-channels are important for generation of LTS (low threshold spike) and rebound bursting during neuronal hyperpolarization (regulation of the neuronal excitability), they can modulate rhythmic activity of the thalamus. We have now clarified this sentence: “Under anesthesia the neuronal hyperpolarization is capable of removing T-channels from inactivation (de-inactivation) and generation of bursting and oscillatory behavior within thalamocortical loop is facilitated.”

Figure 1. Was the frequency artificially cut off at approximately 40 Hz in the upper panel? The imaginary line of the cut off looks artificially straight. If it was not cut off, why has absolutely everything above 40Hz disappeared? The frequency was not artificially cut off but at 40 Hz was under 40 dB in the particular example, we replaced WT heat map on Figure 2.

“While in our study we did not investigated” – line 220

“While in our study we did not investigated the spindle component, we found increased power density in spindle-like frequency range (8–13 Hz)”. Although it is typically 11-16Hz for spindles, perhaps you can say this, however, does it mean that your data contradict the data in [63]? If this is the case, it should be discussed in greater detail. “in mice lacking CaV3.1 channel during quiet awake state and under 1 vol%, but not under 2 vol% of isoflurane anesthesia in CeM”. 2% isoflurane for mice is more than 1 MAC and you describe burst suppression in the next paragraph under 1.4%. What changes would you expect to see in 8-13Hz under 2%? We are referring to the data from our previously published paper (Timic Stamenic, T.; Feseha, S.; Valdez, R.; Zhao, W.; Klawitter, J.; Todorovic, S.M. Alterations in Oscillatory Behavior of Central Medial Thalamic Neurons Demonstrate a Key Role of CaV3.1 Isoform of T-Channels During Isoflurane-Induced Anesthesia. Cereb. Cortex 2019, 29, 1–18, doi:10.1093/cercor/bhz002.), where we found statistically significant difference in relative alpha (8-13 Hz) power under 1% of isoflurane (Figure 8B) but not under 2% of isoflurane (Figure 8C) of isoflurane. We did not analyze spindles. We clarified this in the manuscript.

“Moreover, we found that during inhalation of 2 vol% isoflurane, dominant activity was δ oscillations during suppression mode in the WT mice, but not in the CaV3.1 KO animals”. What was the dominant activity in CaV3.1 KO animals if they have the same suppression to burst ratio as in WT group according to Table 1? In comparison to WT mice, where we observed delta rhythm during suppression in thalamus not in cortex, in CaV3.1 KO animals this was not present. We did not performed analysis but the details can be found in our paper – Timic Stamenic, T.; Feseha, S.; Valdez, R.; Zhao, W.; Klawitter, J.; Todorovic, S.M. Alterations in Oscillatory Behavior of Central Medial Thalamic Neurons Demonstrate a Key Role of CaV3.1 Isoform of T-Channels During Isoflurane-Induced Anesthesia. Cereb. Cortex 2019, 29, 1–18, doi:10.1093/cercor/bhz002

Figure 2 should be accompanied by group data with error bars if you want to show such a big difference between males and females. We published paper with the sex differences under 3β-OH with the appropriate statistical analysis. The Figure 2 (3) presents descriptive thalamic effect of two doses of neurosteroid (hypnotic and anesthetic) and observed sex-dependent effect of the drug. The sex-dependent effect was seen with other neuroactive steroids that we explored in our recent study. Joksimovic, S.M.; Sampath, D.; Krishnan, K.; Covey, D.F.; Jevtovic-Todorovic, V.; Raol, Y.H.; Todorovic, S.M. Differential effects of the novel neurosteroid hypnotic (3β,5β,17β)-3-hydroxyandrostane-17-carbonitrile on electroencephalogram activity in male and female rats. Br. J. Anaesth. 2021, 127, 435–446, doi:10.1016/J.BJA.2021.03.029/ATTACHMENT/B2F1CD94-8A98-4D87-8934-09AB758B4B9B/MMC1.DOCX.

“3.3. Role of the CaV3.3 T-channels in anesthesia induced by isoflurane” It is not clear why only isoflurane was chosen for this discussion. What about sevoflurane and other anesthetics? Unfortunately, to our knowledge there are no data about role of this T-channel isoform and other common anesthetics. This remains an important area of our future investigations.

“The TRN is anatomically positioned as an optimal site to monitor cortical sensory processing and the regulation of the thalamus [90].” This is arguable. Physiologically it sounds very unusual that thalamus monitors cortical processing. I do not understand the mechanism. Activation of the cerebral cortex typically suppresses activity in the thalamus. Additional support/explanaition is advised for this statement. “It has been shown that the local optogenetical TRN activation induces slow wave activity similar to those seen in sleep [91]. Also, during the TRN stimulation animals exhibited behavioral changes consistent with a decrease in arousal state [91].” These studies do not prove the point that thalamus monitors cortical processing. They only show the effect of strong artificial stimulation of TRN on the cortex. We agree and we excluded this section from the manuscript since it was confusing.

“The TRN possess an intrinsic firing ability mediated by CaV3.3 channels expressed predominantly in the dendritic branches that facilitate characteristic bursting induced by T-channel activation during sleep and anesthesia and tonic firing during awake state [89,92].” It would be interesting to discuss the driver for the spontaneous (?) activation of T-channels under these conditions. We excluded this section from the manuscript since it requires extensive discussion that is beyond the scope of the review.

Overall, sections 3.3 and 3.4 seems unfocused (as well as some other sections). They list different studies and results, but it is not clear what the implications are of the findings presented, especially in reference to the purpose of the manuscript. We excluded section 3.3 from the manuscript and added section 3.4 to the previous section (3.2) since it was unfocused.

The conclusions are descriptive and weak. We re-wrote the conclusion section and hope that this reviewer finds it now more suitable.

Round 2

Reviewer 1 Report

This is a much improved manuscript and the authors responses are much appreciated, but there are still many misspellings that must be corrected and could be avoided with a run of an automated spellchecker as noted below:

line 74 chennels

line 77: memrane, alow

line 282: imortant

line 283: detailes

line 297 disappreared

line 298: anestetic

Author Response

The English language editing was done using the editing services listed at
https://www.mdpi.com/authors/english .

Reviewer 2 Report

The authors have done a lot of work in terms of improving this manuscript. Taking into account the difficulty and potential inconclusiveness of the chosen topic itself, I think the current version of the manuscript is as specific as possible. I highly recommend improving the use of English, as there are multiple language errors that should be fixed throughout the manuscript.

Author Response

The English language editing was done using the editing services listed at
https://www.mdpi.com/authors/english .

This manuscript is a resubmission of an earlier submission. The following is a list of the peer review reports and author responses from that submission.